# Robot-Assisted Gait Training in Patients with Multiple Sclerosis: A Randomized Controlled Crossover Trial

**DOI:** 10.3390/medicina57070713

**Published:** 2021-07-14

**Authors:** Cristiano Sconza, Francesco Negrini, Berardo Di Matteo, Alberto Borboni, Gennaro Boccia, Ignas Petrikonis, Edgaras Stankevičius, Roberto Casale

**Affiliations:** 1Department of Biomedical Sciences, Humanitas University, Via Rita Levi Montalcini 4, 20090 Pieve Emanuele, Italy; cristiano.sconza@humanitas.it; 2IRCCS Humanitas Research Hospital, Via Manzoni 56, 20089 Rozzano, Italy; berardo.dimatteo@humanitas.it; 3IRCCS Istituto Ortopedico Galeazzi, 20161 Milan, Italy; francesco.negrini@gmail.com; 4Department of Traumatology, Orthopaedics and Disaster Surgery, First Moscow State Medical University (Sechenov University), 6-1 Bolshaya Pirogovskaya St., 119991 Moscow, Russia; 5Mechanical and Industrial Engineering Department, University of Brescia, 25121 Brescia, Italy; alberto.borboni@unibs.it; 6Department of Clinical and Biological Sciences, University of Turin, 10124 Turin, Italy; gennaro.boccia@unito.it; 7Faculty of Medicine, Medical Academy, Lithuanian University of Health Sciences, 50103 Kaunas, Lithuania; ignas.petrikonis@stud.lsmu.lt; 8Laboratory of Preclinical Drug Investigation, Institute of Cardiology, Lithuanian University of Health Sciences, 50166 Kaunas, Lithuania; 9Rehabilitation and Pain Rehabilitation Unit, Department of High Technology, Habilita Care and Research Hospitals, 24040 Zingonia di Ciserano, Italy; robertocasale@opusmedica.org; 10Opusmedica PC&R, Persons Care & Research Network Piacenza, 29121 Piacenza, Italy

**Keywords:** multiple sclerosis, robotic training, gait recovery, rehabilitation, exercise

## Abstract

*Background and Objectives:* Gait disorders represent one of the most disabling aspects in multiple sclerosis (MS) that strongly influence patient quality of life. The improvement of walking ability is a primary goal for rehabilitation treatment. The aim of this study is to evaluate the effectiveness of robot-assisted gait training (RAGT) in association with physiotherapy treatment in patients affected by MS in comparison with ground conventional gait training. Study design: Randomized controlled crossover trial. *Materials and Methods:* Twenty-seven participants affected by MS with EDSS scores between 3.5 and 7 were enrolled, of whom seventeen completed the study. They received five training sessions per week over five weeks of conventional gait training with (experimental group) or without (control group) the inclusion of RAGT. The patients were prospectively evaluated before and after the first treatment session and, after the crossover phase, before and after the second treatment session. The evaluation was based on the 25-foot walk test (25FW, main outcome), 6 min walk test (6MWT), Tinetti Test, Modified Ashworth Scale, and modified Motricity Index for lower limbs. We also measured disability parameters using Functional Independence Measure and Quality of Life Index, and instrumental kinematic and gait parameters: knee extensor strength, double-time support, step length ratio; 17 patients reached the final evaluation. *Results:* Both groups significantly improved on gait parameters, motor abilities, and autonomy recovery in daily living activities with generally better results of RAGT over control treatment. In particular, the RAGT group improved more than control group in the 25FW (*p* = 0.004) and the 6MWT (*p* = 0.022). *Conclusions:* RAGT is a valid treatment option that in association with physiotherapy could induce positive effects in MS-correlated gait disorders. Our results showed greater effectiveness in recovering gait speed and resistance than conventional gait training.

## 1. Introduction

In Multiple Sclerosis (MS), the highly variable distribution of demyelination areas and axonal loss in the central nervous system (CNS) can lead to very complex and unpredictable neurological deficits and clinical patterns [1]. Gait disorders as reduced speed and stride length, gait asymmetry, increased muscular energy expenditure, balance deficit, and increased risk of falling represent one of the most disabling aspects [2,3,4]. These motor problems strongly influence the level of independence that a person affected by MS is able to achieve, resulting in a severe negative impact on the quality of life [5]. Therefore, the improvement of walking ability is a primary goal for rehabilitation treatment [6]. Many studies demonstrated that a conventional rehabilitation treatment based on physiotherapy could be effective in increasing motor function, gait ability, and patient autonomy [6,7,8,9,10,11,12,13,14]. According to the most recent neurophysiological concepts based on neural plasticity, in recent years the rehabilitative approaches that seem to be more effective in improving functional performance are based on the concept of task-specific repetitive training [15], as in the case of the body weight support treadmill training (BWSTT) for the lower limb rehabilitation [16]. The factors that appear to positively affect the patient outcome are intensity, precocity, repeatability, and specificity in training that incorporates high numbers of task-oriented movement repetitions [17]. Hence, the interest in automated robotic devices for gait training for MS patients has grown. With their consistent, symmetrical lower-limb trajectories, robotic devices provide many proprioceptive inputs that may increase cortical activation and stimulation of central pattern generator (CGPs) in order to improve motor function [18]. The use of Robot Assisted Gait Training (RAGT) allows: repetition of specific and stereotyped movements in order to acquire a correct and reproducible gait pattern in conditions of balance and symmetry, early start of treatment using the activity with body weight support, safeguarding the patient by reducing the fear of falling to increase the quantity and quality of the exercise performed by optimizing the intervention of the therapist [19]. RAGT has been applied to several neurological pathologies such as stroke, traumatic brain injury (TBI), spinal cord injury (SCI), MS, Parkinson’s disease and cerebral palsy, suggesting improvements in functional performance and gait parameters [20]. Namely, a 2018 pilot study showed that RAGT seems to have a beneficial effect on lower limb strength and autonomy in walking in MS patients [21]. A recent review and meta-analysis suggest that RAGT is not only safe and tolerated, but even superior to conventional gait training in improving perceived fatigue and spasticity [22]. Furthermore, RAGT used in combination with other new technologies seems to be highly tolerated and supports commitment and motivation in MS patients [23], a very important aspect to consider in such a chronic progressive pathology that needs continuous effort by the patient, often for many years. However, to this date, there are no certain and reliable data on the long-term effect of RAGT. The aim of this randomized and controlled crossover study is to evaluate the effectiveness of gait training using a robot-driven gait orthosis in patients affected by MS in comparison with ground conventional gait training. The changes in gait pattern, motor ability and autonomy in the functional activities of daily living will be assessed by using validated clinical and functional scales and quantitative instrumental analysis of gait kinematic parameters.

## 2. Materials and Methods

A randomized controlled crossover trial was conducted. All subjects provided written informed consent prior to being enrolled and procedures were conducted according to the Declaration of Helsinki. The protocol (LK.S P07) was approved by the Local Ethical Committee of Bergamo, Italy, and is registered with ClinicalTrials.gov (NCT02291107).

Population: We enrolled twenty-seven patients affected by MS and admitted to our Department of Physical and Rehabilitation Medicine from June 2014 to December 2015. Eight of them were excluded for different reasons, six because they did not meet inclusion criteria, and two because they declined to participate. The remaining nineteen participants (15 women and 4 men, aged 36 to 74 years old) were admitted to the study. During the study, two patients dropped out due to major exacerbation of the symptoms of the disease and consequent inability to continue treatment. The final total sample was seventeen patients. Each patient was subjected to a physical and subjective examination performed by a physician experienced in neurorehabilitation conditions in order to evaluate the inclusion and exclusion criteria. To be included in the study, the participants needed to have: (1) a diagnosis of MS made by a neurologist according to the McDonald’s Criteria [24], (2) a stable phase of disease for at least 3 months, and (3) the ability to walk 25 foots without assistance, (4) an Expanded Disability Status Scale (EDSS) score between 3.5 and 7 [25]. The exclusion criteria were (1) exacerbation of the disease in the last 3 months, (2) unstable medical disorders (recent myocardial infarction, uncontrolled hypertension, diabetes), (3) symptomatic fall in blood pressure when standing, (4) vascular claudication, (5) severe–moderate cognitive impairment (Mini Mental State ≤ 21), or (6) other neurological conditions in addition to MS (stroke, TBI, low back pain, peripheral nerve disease). Besides, (7) patients that could not use the RAGT for the following reasons were excluded: (i) body weight greater than 135 kg, (ii) height more than 200 cm, (iii) limb-length discrepancy greater than 2 cm, (iv) joint problems (hip or leg) that strongly limited range of motion or cause important pain with movement, (v) unstable fractures, (vi) pressure sores with any skin lesion in areas in contact with the body harness or the robot-driven gait orthotic apparatus.

The participants enrolled in the study were stratified according to the form of the disease (relapsing-remitting (RRMS), progressive relapsing (PRMS), secondary progressive (SPMS) and primary progressive (PPMS)) and the degree of disability (by EDSS scores from 3.5 to 4.5 (group 1), from 5 to 6 (group 2), from 6.5 to 7 (group 3)).

Protocol: After the completion of all baseline measurements (T0), the patients were randomly assigned by an external assistant into one of two treatments following strictly a sequential order: an experimental treatment group A or a control treatment group B (Figure 1). The randomization list was generated by an independent statistician using the Schrage algorithm [26], and kept in a dedicated office. In particular, sealed envelopes numbered progressively were used containing the assignment of the treatment (group A or B). The therapist contacted the office just before performing the treatment to know the patient allocation; the envelope was then opened to determine the allocation group and the patients included in the randomization list.

Treatment group A performed a rehabilitation treatment lasting 1 h and 30 min of walk training on the Lokomat^®^ plus a physiotherapy session consisting of a general exercise program and ground gait training; control group B performed only the physiotherapy treatment for 1 h and 30 min. The first protocol phase ended after 25 training sessions for both treatment groups, 5 days per week, for 5 weeks (T1). On completion of the first phase and following a 4-month washout period (T2), the subjects were crossed-over to the alternate treatment. As stated, both treatment groups concluded after 25 therapy sessions with the same cadence (T3). The end of the second randomized phase marked the end of the study. Each patient underwent a total of 50 treatment sessions and participated equally in both rehabilitation protocols. During the washout period, patients were allowed to independently reproduce physical exercises at home. They were required to refrain from further treatment with the physiotherapist or through medical (i.e., botulinum toxin) or instrumental therapies during the entire washout period.

All patients were evaluated by a blinded observer using outcome tests before (T0) and after (T1) the first treatment phase and before (T2) and after (T3) the second.

We performed a standard assessment procedure on all patients recruited using scales of proven reliability, validity, and sensitivity to change. An independent physician, not involved in the rehabilitation process, performed the assessments. Therapists blinded to the aim of the study performed the treatments. Data analysis was performed offline by an external assessor unaware of the randomization process. The same assessor that took the pre-treatment measurement and who remained blinded to the treatment assignment to the subjects performed the post-treatment assessments.

Sample Size: The sample size was estimated according to the expression below:(1)$$n\cdot b=\frac{4\sigma^{2}\;{\left(z_{crit}+z_{pwr}\right)}^{2}}{D^{2}}$$
where: *n* is the sample size, *b* is the number of the arms and is equal to 2, *σ* is the estimated standard deviation for both groups (experimental and control) and is taken from reference [27], *D* is the minimum expected difference and it is taken on the value obtained in reference [27], *z_crit_* is assumed to be 1960 and *z_pwr_* is assumed to be 1.645. The a priori calculation of the sample size was made on the primary outcome variable: the timed 25-foot walk (25FW), resulting in 17.

Instruments: The RAGT used is a robotic device set up as an exoskeleton on the lower limbs of the patient (Lokomat^®^, Hocoma) [28]. The system uses a dynamic body weight-support system to support the participant above a motorized treadmill that is synchronized with the gait orthosis in a virtual reality environment with constant audio and visual feedback used to perform various tasks.

Experimental Treatment for Group A: The participants received 25 sessions of robotically driven gait orthosis training on the Lokomat^®^. Training occurred 5 days/week for 5 weeks and each training session on the Lokomat^®^ lasted 30 min. All sessions were supervised by a trained therapist. All participants started with 40% body weight support and an initial treadmill speed of 1.5 km/h. Body weight-support was used primarily to facilitate an increase in walking speed; therefore, progression of training across subsequent sessions was standardized by preferentially increasing speed and then unloading body weight-support. Speed was increased to a range of 2.2 to 2.5 km/h before body weight-support was decreased. An active attempt to enhance the level of training at each session was allowed according to Lo [29]. After every Lokomat^®^ session, the participants also performed 60 min of physiotherapy including a general exercise program and ground gait training.

Control Treatment for group B: The participants received 25 sessions of conventional physiotherapy treatment. Training occurred 5 days/week for 5 weeks, and each training session lasted 1 h and half. The conventional physiotherapy treatment consisted of a general exercise program and ground gait training. The general exercise program consisted of cardiovascular warm-up exercises, muscle stretching exercises, active-assisted or active isometric and isotonic exercises for the main muscles of the trunk and limbs, relaxation exercises, coordination and static/dynamic balance exercises. The conventional gait therapy was based on the proprioceptive neuromuscular facilitation concept, training in walking on different surfaces with or without appropriate walking aids, exercises for the restoration of a correct gait pattern, implementation of residual compensatory strategies and progressive increase in walking resistance. The same trained therapist treated all the patients in this group and standardized the duration of each part of the treatment.

Outcome Variables:

Primary Outcome Measures: Timed 25-foot walk (25FW) for the gait speed assessment. It is the first component of the Multiple Sclerosis Functional Composite (MSCF) for the study and measure of quantitative mobility and leg function performance in MS clinical trials [30].

Secondary Outcome Measures: Lower limb motor and function skills: Modified Motricity Index (MI) for the assessment of lower limb motor function [31] and Modified Ashworth Scale (ASH) to measure lower limb spasticity [31].

Gait and balance skills: Timed 10 m walking test (TWT) for the gait speed assessment [32]; 6 min walking test (6MWT) to measure gait performance in terms of endurance [33] and Tinetti Test (TT) for qualitative measures of balance and gait [34,35].

Instrumental kinematic parameters: Knee extensor strength (KP) for the assessment of knee extensor strength by dynamometer measurement [36]; double time support (DST) that is a kinematic parameter corresponding to the duration of the double support phase of a gait cycle, calculated as ms/% [27] and step length ratio (SLR) that is a kinematic parameter corresponding to gait symmetry, calculated as the ratio between the step length of both legs (shorter step length/longer step length) [27].

Disability and Quality of Life: EDSS for the assessment of disability in multiple sclerosis patients [25]; Functional Independence Measure (FIM) to measure autonomy during daily activities [37] and Quality of Life Index (QL Index–SF36) to measure health-related quality of life [38].

Statistical Analysis: Firstly, to test whether the phase of washout was effective, the baselines of the two sequences (T0 and T2) were compared with paired t tests or Wilcoxon’s tests accordingly to the results of the normality tests. After that, we tested the differences between experimental and control treatments by a series of analyses of covariance for each outcome with group as the main effect (experimental vs. control) and the baseline value, gender, age, and EDSS scores at baseline as covariates. To measure changes in outcomes, the mean change from baseline was calculated for each treatment phase. Significance was recognized when *p* < 0.05. The effect size (ES) was calculated as post-training mean minus pre-training mean divided by pooled standard deviation of pre- and post-training [39]. Threshold values for effect size statistics were: <0.2, trivial; >0.2, small; >0.5, medium; >0.8 large; >1.3, very large [40]. To compare changes between treatment groups (inter-participant: Sequence A vs. Sequence B at each phase) in the primary outcome measures (25FW), two sample t tests were used. The data were analyzed using the SPSS version 19 software (SPSS Inc., Chicago, IL, USA) following an intention-to-treat (ITT) and a last-observation-carried-forward (LOCF) method. Thus, all patients treated, including those who dropped out of the experiment, were considered.

## 3. Results

The characteristics of the patients included in the study are presented in Table 1. Regarding the first phase of the protocol, non-significant differences between T0 and T2 showed that the washout phase was effective for all outcomes (all *p* values > 0.12). The forest plot in Figure 2 shows an overall advantage of RAGT over control treatment for most outcomes. Despite such a trend, when the changes at POST-Treatment T3 were controlled for the baseline values (PRE-Treatment T2), the only outcomes that resulted in a significant main effect for the treatment were the 25FW (*p* = 0.004, Figure 3 left panel) and the 6MWT (*p* = 0.022, Figure 3 right panel).

Figure 3 shows the individual changes in the 25FW test from baseline (T2). There was an overall trend to a greater improvement of 25FW outcomes in the second phase for both experimental (*p* = 0.01) and control treatment (*p* = 0.05). In Table 2, the complete results of the analysis of covariance are reported.

## 4. Discussion

The results of our study show that rehabilitation treatment using an approach that combines conventional physiotherapy and RAGT is capable of increasing gait motor capacity and kinematic parameters while overall reducing disability in MS patients. The most relevant result of our work is that this type of treatment approach can significantly improve patients’ performance in terms of gait speed and endurance compared with conventional ground training. The post-treatment evaluation of lower limb motor parameters has shown an increase in mobility and reduction in muscle spasticity in both groups with a higher efficacy obtained by RAGT, supporting the results of a recent meta-analysis on the topic [22]. The average improvement in 25FW (14.9%) in the intervention group was larger than the minimal clinical important difference [41], therefore suggesting a clinically relevant enhancement of mobility. Similar results were found in the evaluation of gait and balance skills, with patients in both groups experiencing significant improvements in postural control, knee extension strength and gait kinematic parameters as the reduction in double time support and increase in walking symmetry, especially in the RAGT group. The change in intensity load on the femoral quadriceps was also measured with similar data published by Beer et al. [36], thus confirming a moderate to large improvement in gait speed and endurance as well as an increase in knee extensor strength in the group of patients with severe disability (EDSS of 6–7.5). The assumption was that the Lokomat^®^ could create a higher intensity eccentric quadriceps exercise differently from what occurs during ground training where muscle activation is mainly concentric and of lower intensity [21,42]. Interestingly, the parameters assessing disability and perception of quality of life both improved after treatments in all study groups with a higher positive trend in all outcomes measured in the RAGT than in the control group. The only exception observed was in the evaluation of the patients’ emotional-mental health perception. The above results are consistent with the evidence found in the scientific literature [8,9,10,11,12,13], which confirms the effectiveness of the physiotherapy approach in the treatment of MS patients.

There are already studies in the literature that have investigated the effects of RAGT on MS patients and have shown encouraging results regarding improvements in functional status, gait performance and quality of life [17,20,21,22,27,36,42,43,44,45,46,47,48,49,50,51,52,53,54,55]. Most of them used different RAGT protocols in terms of treatment sessions, frequency, and criteria for progression as well as a different study design and methods making data poorly comparable [27,44]. To overcome this difficulty in comparing different results and to reduce the confounding clinical heterogeneity that exists among MS patients, a crossover design was chosen in our protocol as suggested by other authors [27]. The advantage of this type of study design is that having each participant as its own control would allow for a more accurate interpretation of the treatment effects on each patient. Another controversial issue we tried to minimize was the length of the washout period. In our study, we decided to set a long washout time of 16 weeks (4 months) to avoid any carryover effect. This time interval proved to be correct and we did not experience any carryover effects after the washout period. Unfortunately, this result showed at the same time that the beneficial effects obtained by RAGT and/or physiotherapy training were of limited duration. This observation is an issue of concern and will be an issue of further research in the future [55]. Moreover, in order to have comparative results with other studies, our treatment protocol consisted of 25 sessions administered five times per week, mirroring the one used by Beer et al. [36] and showing similar results. A major problem in the assessment of the efficacy of all form of rehabilitation is related to the patient’s level of disability, with data suggesting that also RAGT effectiveness may be related to the degree of disability [45]. Indeed, the use of RAGT has been emphasized in relation to more severely affected patients who would require greater effort on the part of physiotherapists [44], the most important reason for the use of robotic devices. In our study, we assessed the efficacy of RAGT in a group of patients with an EDSS between 3.5 and 7, and we believe that extrapolation to a more severe form of MS could be not correct and that further studies on different population settings are mandatory. With regard to outcome variables concerning the patients’ perception of their own physical health state, evaluated by the SF36 scale, a significant improvement was seen in our study, similarly to that seen by other authors [51] comparing RAGT versus BWSTT as well as versus conventional treatments. In a study [47] comparing RAGT with sensory integration balance training (SIBT), significant results were shown in both groups in terms of improved balance and postural control, more pronounced in the SIBT group, as well as an increase in walking speed in patients undergoing RAGT. An important difference in this study was the use of an end-effector type robotic device rather than the Lokomat^®^. This raised the general question of how comparable the results obtained from rehabilitation training conducted with different robotic devices were. There is currently no definitive answer in the literature. The results reported by reviews and meta-analyses [22,53] show significant differences between studies mainly on treatment protocols, follow-up period, and characteristics of the analyzed sample, often small and composed of patients with different types of MS and various degrees of disability.

A complete understanding of the benefits of using these new technologies in rehabilitation is limited and many questions remain without a clear answer. In the context of real-world rehabilitation, the criticalities are many and related to the applicability of robotic devices within the rehabilitation treatment processes currently in use. The timing in planning RAGT within a rehabilitation program, the potential interaction of specific and different devices with the patient’s functional profile are of great interest to both clinicians and researchers. Another important issue not sufficiently considered is the impact of different technologies on the organization of the rehabilitation team which would require specific training and how much this can influence the results.

The use of RAGT certainly entails significant costs linked to the purchase and management of the exoskeleton; on the other hand, it allows the number of physiotherapists directly engaged with the patient to be reduced: one physiotherapist is able to manage the treatment with the exoskeleton, whereas two operators are usually required to perform gait training. However, our study was not focused on performing a comparative cost–benefit analysis, but only to evaluate the effectiveness of two different treatment strategies. Nevertheless, this aspect is relevant in terms of health policies and requires further investigations to establish if RAGT might be economically rewarding.

Although in our study we tried to overcome some critical points described so far and the data highlight some interesting results, some limitations are still present. Even if the sample size of patients was statistically correct to achieve a significant and robust data analysis of the primary outcome (25FW), more data are needed. In addition, stratification of patients by disease type and degree of disability reduced the sample size. Furthermore, it was not possible to evaluate a potential correlation between the prevalence of a specific clinical symptom of cerebellar, sensory, or motor origin and the outcomes studied. Nor was it possible to compare data from other studies in which patients with severe gait disability achieved greater benefits from RAGT than others; in fact, our study showed a significant improvement in outcome measurements regardless of the patient’s EDSS classification. For these reasons, the future goal would be to increase the sample size in order to obtain more data with greater statistical power and results more closely reflecting reality. Although our study did not perform post-treatment follow-up per se, due to the rather long washout interval (4 months between T1–T2) it is possible to state that the effectiveness of the treatments was remarkably reduced in most patients. However, we were not able to draw firm conclusions about the possible long-term effects of the treatment.

## 5. Conclusions

Our study, along with an extensive critical review of the scientific literature, draws attention to several fundamental points and questions, which must be considered in the rehabilitation process of MS patients. Motor outcomes, particularly those related to increased walking speed and endurance, seem to derive important benefits from the use of RAGT in combination with conventional physiotherapy, but it is necessary to understand how long these tools positively influence those functions and the patient’s participation in social life. The intensity of the treatment seems to be a fundamental factor for the improvement of the outcomes but of the utmost importance is the ability to detect the correct type of patients who can greatly benefit from robot-assisted rehabilitation.

## Figures and Tables

**Figure 1 medicina-57-00713-f001:**
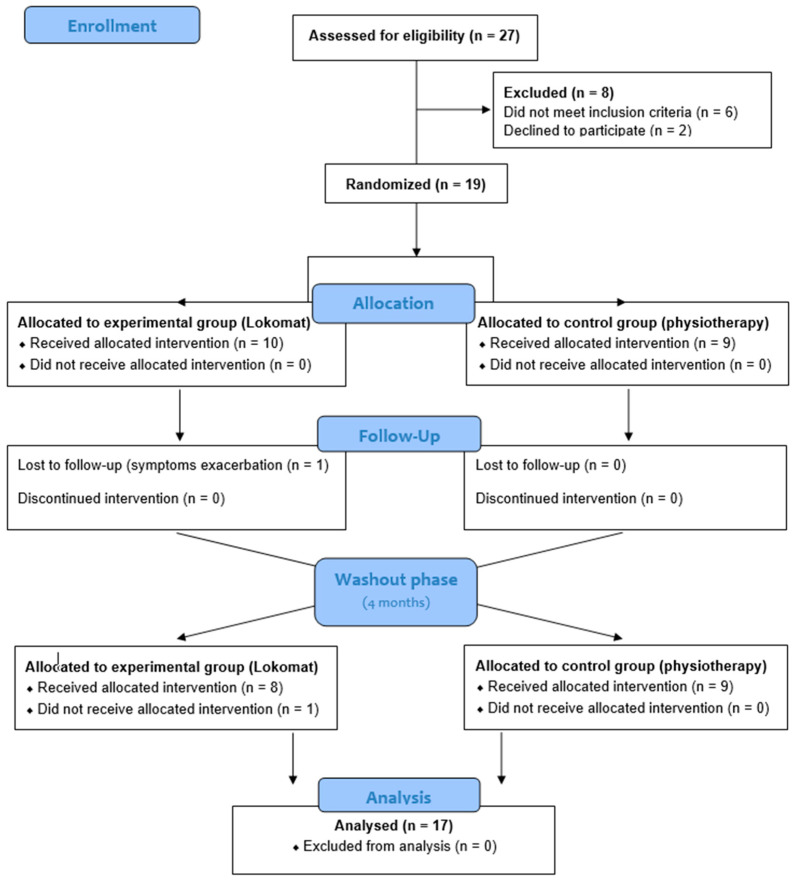
CONSORT flow diagram for the crossover randomized trial.

**Figure 2 medicina-57-00713-f002:**
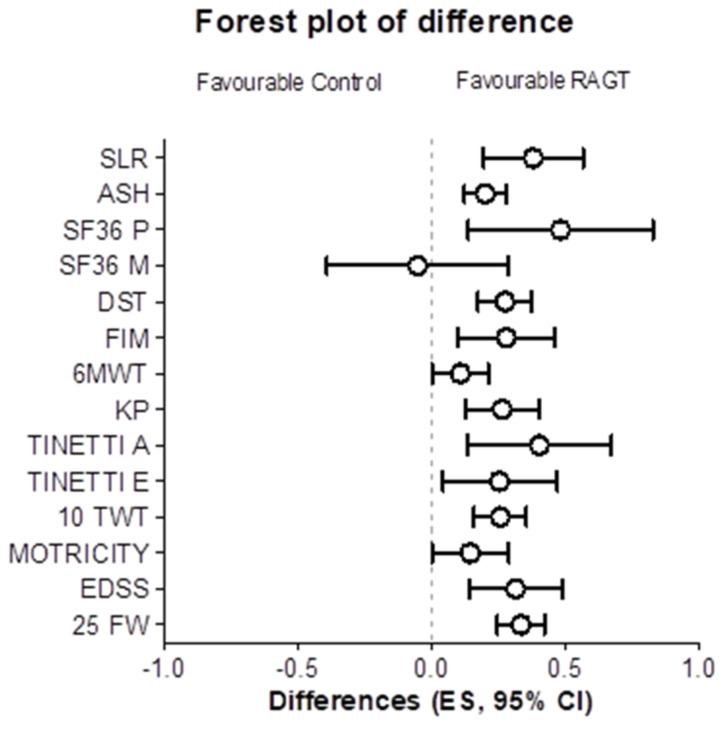
Forest plot of the treatment differences. The effect sizes (ES) were calculated as changes from pre-training for each intervention and then the ES differences between robot assisted gait training (RAGT) and control intervention were computed to create the forest plot. The quantitative report of the ES difference is reported Table 2.

**Figure 3 medicina-57-00713-f003:**
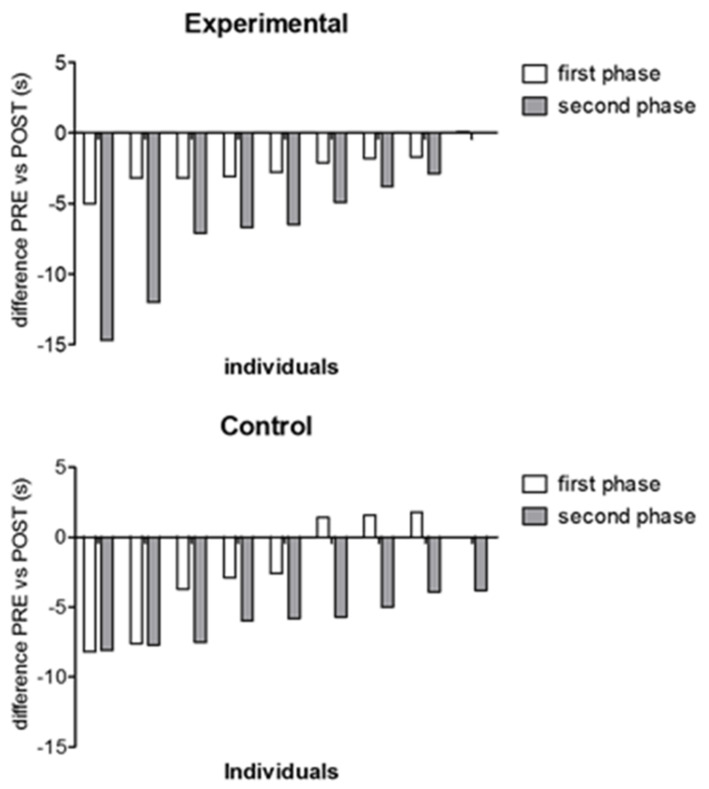
Individual changes from baseline in walking execution time. Individual changes from baseline the 25FW test are reported separated in the two phases for experimental (upper panel) and control (lower panel) treatment. The values are sorted from the greater to the lower individual change. There was an overall trend to greater changes in the second phase (from T2 to T3) than in the first (from T0 to T1) for both experimental (*p* = 0.01) and control treatment (*p* = 0.05).

**Table 1 medicina-57-00713-t001:** Characteristics of patients recruited in the study.

Patient Characteristics	(*N*)
Gender	
Female	16
Male	3
Type of Multiple Sclerosis	
Relapsing Remitting	7
Progressive Relapsing	4
Secondary Progressive	5
Primary Progressive	3
EDSS	
3.5–4.5	5
5–6	7
6.5–7	7

EDSS: Expanded Disability Status Scale.

**Table 2 medicina-57-00713-t002:** Outcome changes according to the training intervention.

	PRE (T2)	POST (T3)	Δ%	ES	ES Interpretation	ES Difference (95% Confidence Intervals)	Main Treatment Effects		
							F	*p*	Partial η^2^
25FW									
RAGT	45.3 ± 12.8	38.7 ± 11.6 ***	−14.9	−0.52	Medium	0.33 (0.24–0.42)	13.4	0.004	0.549
Control	43.0 ± 13.7	40.5 ± 12.9 **	−5.7	−0.19	Trivial				
EDSS									
RAGT	5.6 ± 1.0	5.2 ± 1.0 ***	−7.2	−0.39	Small	0.31 (0.14–0.48)	0.02	0.892	0.002
Control	5.5 ± 1.1	5.4 ± 1.0	−1.4	−0.08	Trivial				
Motricity									
RAGT	114 ± 27	124 ± 30 *	9	0.35	Small	0.15 (0.01–0.28)	3.0	0.108	0.218
Control	113 ± 26	119 ± 27 **	5	0.20	Small				
10 TWT									
RAGT	0.45 ± 0.19	0.55 ± 0.21 ***	23.2	0.46	Small	0.26 (0.15–0.35)	0.965	0.347	0.081
Control	0.49 ± 0.24	0.54 ± 0.24 ***	11.8	0.20	Small				
Tinetti E									
RAGT	9.11 ± 3.40	11.41 ± 3.84 ***	28.0	0.61	Medium	0.26 (0.04–0.46)	0.313	0.587	0.028
Control	9.35 ± 3.46	10.64 ± 3.83 **	14.4	0.35	Small				
Tinetti A									
RAGT	6.05 ± 2.41	8.05 ± 2.33 **	46.4	0.78	Medium	0.41 (0.13–0.67)	0.01	0.981	0.001
Control	6.58 ± 2.55	7.52 ± 2.42 *	21.9	0.37	Small				
SF36 P									
RAGT	30.4 ± 6.7	35.7 ± 7.3 *	19.6	0.70	Medium	0.48 (−0.38–0.28)	0.02	0.873	0.002
Control	31.7 ± 8.3	33.5 ± 8.1	7.2	0.22	Small				
SF36 M									
RAGT	48.6 ± 13.7	49.3 ± 13.7	2.8	0.05	Trivial	−0.05 (−0.38–0.28)	2.6	0.134	0.192
Control	47.3 ± 13.7	48.6 ± 12.2	5.1	0.10	Trivial				
KP									
RAGT	18.3 ± 11.4	21.8 ± 13.6 **	18.5	0.31	Small	0.30 (0.17–0.44)	0.001	0.974	<0.001
Control	20.0 ± 12.7	20.1 ± 13.0	0.2	0.01	Trivial				
6MWT									
RAGT	137 ± 84	158 ± 89 ***	19.2	0.24	Small	0.11 (0.01–0.21)	7.13	0.022	0.393
Control	146 ± 95	159 ± 98 **	10.6	0.13	Trivial				
FIM									
RAGT	101.2 ± 14.7	106.9 ± 12.5 **	6.0	0.41	Small	0.28 (0.10–0.46)	3.0	0.110	0.216
Control	103.0 ± 13.1	104.7 ± 13.3 *	1.6	0.13	Trivial				
DST									
RAGT	42.5 ± 10.8	39.2 ± 12.6 *	−9.1	−0.28	Small	0.27 (0.17–0.37)	0.595	0.457	0.051
Control	42.0 ± 10.9	42.0 ± 12.2	−0.8	−0.01	Trivial				
ASH									
RAGT	4.6 ± 4.4	3.3 ± 2.9 *	−25	−0.34	Small	0.20 (0.03–0.36)	0.158	0.698	0.014
Control	4.2 ± 4.55	3.7 ± 3.6	−11	−0.14	Trivial				
SLR									
RAGT	0.86 ± 0.06	0.91 ± 0.05 **	6.5	0.83	Large	0.38 (−0.01–0.77)	1.1	0.304	0.096
Control	0.87 ± 0.06	0.90 ± 0.06 *	3.6	0.45	Small				

Δ% = percent difference with respect to pre-training; ES = effect size; Partial η^2^ = partial eta-squared. Values are expressed as means and standard deviation. * *p* < 0.05; ** *p* < 0.01; *** *p* < 0.001. When the main effect of ANCOVA resulted significant, the *p* values are reported in bold case to highlight the different responses to the training interventions. 25FW: 25-Foot Walk, EDSS: Expanded Disability Status Scale, 10 TWT: 10 m walking test, SF36: Short Form 36, KP: Knee extensor strength, 6MWT: 6 min walking test, FIM: Functional Independence Measure, DST: double time support, ASH: Modified Ashworth Scale, SLR: step length ratio.

## Data Availability

The data presented in this study are available on request from the corresponding author. The data are not publicly available due to privacy and ethical restrictions.

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
