# Peer review of "Robot-Assisted Gait Training in Patients with Multiple Sclerosis: A Randomized Controlled Crossover Trial"

_medicina, 2021, doi:10.3390/medicina57070713_

Round 1

Reviewer 1 Report

This study by Sconza et al is a crossover RCT. It revealed more benefits of robot-assisted gait training (RAGT) than the conventional gait training in patients with MS. Generally speaking, this is a well-structured study and has direct clinical application. There are some questions needed to be revised as below:

  1. You mentioned this is a double-blinded RCT (line106), but how to “double-blinded” to participants and treatment-giver of these 2 different training?
  2. Did the allocation conceal during the randomization process? If yes, please describe that.
  3. Why did you decide the washout period as “4 weeks”? Is there any reference? And did all subjects receive any training or treatment during this period? If yes, please also describe that and to see if affect the outcome.
  4. Line 272: “SLR p= 787“. Was that incorrect? Should it be 0.787?
  5. Line 272-279 : “Despite such a trend… and the 6MWT (p = 0.022, Figure 3 right panel).”  Could you explain this sentence more clearly?
  6. Could you add the number of mean effect and 95% CI on Figure 2?

Author Response

We appreciate very much the editor and the reviewers for the constructive comments. We also

thank the editor and the reviewers for the effort and time put into the review of the manuscript.

Each comment has been carefully considered point by point and responded. We have highlighted

the changes within the manuscript.

Here is a point-by-point response to the reviewers’ comments and concerns.

We also performed a review of the English language and style trying to improve written expression and comprehension. We have adapted the bibliography according to the changes made.

Comment 1: You mentioned this is a double-blinded RCT (line106), but how to “double-blinded” to participants and treatment-giver of these 2 different training

Response:

Thank you for your precise comment. We mentioned this is a double-blinded RCT because the physician that performed the pre-and post-treatment assessments and the external assessor that analyzed data were blinded to the patient’s treatment. As you rightly pointed out, it was not possible to keep the patient blinded too. We changed the title and the blinding methods in the text.

Comment 2: Did the allocation conceal during the randomization process? If yes, please describe that

Response:                                                   

Thanks for your comment; we have edited the text based on your valuable comment as follows.

“The randomization list was generated by an independent statistician using the Schrage algorithm26, and kept in a dedicated office. In particular, sealed envelopes numbered progressively were used containing the assignment of the treatment (group A or B). The therapist contacted the office just before performing the treatment to know the patient allocation; the envelope was then opened to determine the allocation group and the patients included in the randomization list.”

Comment 3: Why did you decide the washout period as “4 weeks”? Is there any reference? And did all subjects receive any training or treatment during this period? If yes, please also describe that and to see if affect the outcome.

Response:

Thanks for your comment. In our study, we set a long washout period of 16 weeks (4 months). We decided this because a previous study on this topic (Lo 2008) showed that 6 weeks was insufficient for outcome measurements to fully return to baseline. Based on this result, the washout time in our study was doubled and set at 4 months to avoid any carry-over effects.

We also included in the text what the patients were allowed to do in this period: “During the washout period, patients were allowed to independently reproduce physical exercises at home. They were required to refrain from further treatment with the physiotherapist or through medical (i.e. botulinum toxin) or instrumental therapies during the entire washout period.”

Comment 4: Line 272: “SLR p= 787“. Was that incorrect? Should it be 0.787?

Response:

Thank you for your precise comment. We have edited the text correctly.

Comment 5: Line 272-279 : “Despite such a trend… and the 6MWT (p = 0.022, Figure 3 right panel).”  Could you explain this sentence more clearly?

Response:

We agree with the reviewer that the sentence was not clear. We rephrased as follows: “Despite such a trend, when the changes at POST-Treatment T3 were controlled for the baseline values (PRE-Treatment T2), the only outcomes that resulted in a significant main effect for the treatment were the 25FW (p = 0.004, Figure 3 left panel) and the 6MWT (p = 0.022, Figure 3 right panel).”

Comment 6: Could you add the number of mean effect and 95% CI on Figure 2?

Response:

Thanks for asking to increase the informativeness of the figure 2. As including the numbers in the figure resulted in a too much busy figure, to address the reviewer’s request we added the 95%CI in the table2 so that the reader have all the quantitative data needed to understand the difference between the experimental and the control group. We specified this in the figure 2 caption.

Reviewer 2 Report

Thank you for the opportunity to review this paper. It is very interesting and well-written. Some minor edits and additions would benefit the final version. I believe it would be of interest to those from a clinical and robotics background. I have some comments below and some suggestions that, respectfully, I feel would improve the paper. The sample size is small, but has been estimated. The number of clinical outcome measures should be praised, and the engagement of the participants. From a clinical perspective, I would like to see this emphasised more so in the discussion. I have provided some notes below to guidance this. 

Line and Comment:

32-33: “They received 5 training sessions/week…” is a little confusing. Does this mean 5 session per week, or 5 training sessions over 5 weeks, or something else. Also, the wording “plus and without” is nonstandard, please rephrase.

43: what does “overall advantage” mean?

43: “significant main effect for the RAGT group” meaning it was significantly better/improved?

73: “…repetitions (add a reference)”

101: Change “The improvements…” to “The changes…”. Improvements are suggestive of already knowing there will be improvements when there may not be.

112: “SM” or “MS”?

31: Twenty-seven participants are enrolled; however, 19 started the study, 2 dropped out, leaving 17. I would suggest a reword in the abstract to include the number of participants that completed the study. Stating 27 in the abstract is not clear and may mislead the reader.

117: include the number of males.

147: how were they randomly assigned? Provide references to the methods where possible.

115: reference of ‘Schrage Algorithm’.

153: what was the “conventional rehabilitation treatment”? (I see this is discussed further in the paper – perhaps comment on this for the reader)

158: “previously…stated/reported”

180: Report the sample size as ‘a priori’ or ‘post hoc’. I would assume a priori, but best to be clear.

183: an image/figure of the exoskeleton would be of interest to the reader, or a link to where this can be viewed.   

269-272: The is a small section of the written text dedicated to the non-significant differences and their p-values, can this be included in a table somewhere? Or, are these important to report at this level of detail.

291: report the qualitative thresholds for the chosen effect size (i.e. small, medium, large), with a reference.

Table 1: What is the 3rd Category – Average Type of Multiple Sclerosis = 59 = 50.4%. What is this? A reader would not know what 50.4% means.

Figure 3: does this figure provide the reader with anything beyond the written report?

General comments:

114: This project was completed 6 year ago(?). Has the introduction/literature been updated since this project to reflect the most up to date evidence?

Data analysis and interpretation: From a clinical perspective, some interpretation of the results in terms of the clinical importance/significance of the noted changes would be of interest. The concept of the ‘smallest worthwhile change’ or ‘minimal clinically significant difference (MCID)’ could be applied for offer this. Based on the number of clinical outcome measures reported – which should be praised – the MCID or alike would offer a clinically meaningful assessment of these findings, in addition to the common/standard reliance of p-values. Additionally, some cost-based assessment could be of interest, i.e. what was the cost of delivering both interventions - clinicians would be interested in this. However, I will default to the journal editor to decide of this comment is in keeping with the Journal and its target audience.

Author Response

We appreciate very much the editor and the reviewers for the constructive comments. We also

thank the editor and the reviewers for the effort and time put into the review of the manuscript.

Each comment has been carefully considered point by point and responded. We have highlighted

the changes within the manuscript.

Here is a point-by-point response to the reviewers’ comments and concerns.

We also performed a review of the English language and style trying to improve written expression and comprehension. We have adapted the bibliography according to the changes made.

Manuscript number: medicina-1249762

Title: Robot-assisted Gait Training in Patients with Multiple Sclerosis: A Double-blind Randomized Controlled Crossover Trial

Comment 1: 32-33: “They received 5 training sessions/week…” is a little confusing. Does this mean 5 session per week, or 5 training sessions over 5 weeks, or something else. Also, the wording “plus and without” is nonstandard, please rephrase.

Response:

We agree with the reviewer that the sentence was confusing. We rephrased as follows: “They received five training sessions per week over five weeks of conventional gait training with (experimental group) or without (control group) the inclusion of RAGT.”

Comment 2:  43: what does “overall advantage” mean?

Response:

We agree with the reviewer that the sentence was confusing. We rephrased as follows: “… with generally better results of…”

Comment 3: 43: “significant main effect for the RAGT group” meaning it was significantly better/improved?

Response:

Again we agree with the reviewer that the sentence was confusing. We rephrased as follows: “In particular, RAGT group improved more than control group in the 25FW (p = 0.004) and the 6MWT (p = 0.022).”

Comment 4: 73: “…repetitions (add a reference)”

Response:

Thank you for the comment. We added the reference in the text. “Patt N, Kool J, Hersche R, Oberste M, Walzik D, Joisten N, Caminada D, Ferrara F, Gonzenbach R, Nigg CR, Kamm CP, Zimmer P, Bansi J. High-intensity interval training and energy management education, compared with moderate continuous training and progressive muscle relaxation, for improving health-related quality of life in persons with multiple sclerosis: study protocol of a randomized con-trolled superiority trial with six months' follow-up. BMC Neurol. 2021 Feb 11;21(1):65. doi: 10.1186/s12883-021-02084-0. PMID: 33573608; PMCID: PMC7877079.”

Comment 5: 101: Change “The improvements…” to “The changes…”. Improvements are suggestive of already knowing there will be improvements when there may not be.

Response:

Thank you for your precise comments. We have corrected the text.

Comment 6: 112: “SM” or “MS”?

Response:

Thank you for your precise comments. We have edited the text accordingly.

Comment 7: 31: Twenty-seven participants are enrolled; however, 19 started the study, 2 dropped out, leaving 17. I would suggest a reword in the abstract to include the number of participants that completed the study. Stating 27 in the abstract is not clear and may mislead the reader.

Response:

We agree with the reviewer that the sentence was confusing. We rephrased as follows: “Twenty-seven participants affected by MS with EDSS scores between 3.5 and 7 were enrolled, of whom seventeen completed the study.”

Comment 8: 117 include the number of males.

Response:

Thank you for your precise comments. We have edited the text accordingly.

Comment 9: 147: how were they randomly assigned? Provide references to the methods where possible.

Response:

Thank you for your accurate comment. We have modified the text as follows in order to clarify the methods of randomization and allocation: “After the completion of all baseline measurements (T0), the patients were randomly assigned by an external assistant into one of two treat-ments following strictly a sequential order: an experimental treatment group A or a control treatment group B (Figure 1). The randomization list was generated by an independent statistician using the Schrage algorithm26, and kept in a dedicated office. In particular, sealed envelopes numbered progressively were used containing the assignment of the treatment (group A or B). The therapist contacted the office just before performing the treatment to know the patient allocation; the envelope was then opened to determine the allocation group and the patients included in the randomization list.”

Comment 10: 150: reference of ‘Schrage Algorithm’.

Response:

Thank you for the comment. We added the reference in the text. “Schrage, L. A More Portable FORTRAN Random Number Generator. ACM Transactions on Mathemati-cal Software 5 (2): 132-138; 1979.         21b Riener, R.Technology of the robotic gait orthosis Loko-mat (2016) Neurorehabilitation Technology, Second Edition, pp. 395-407.”

Comment 11: 153: what was the “conventional rehabilitation treatment”? (I see this is discussed further in the paper – perhaps comment on this for the reader)

Response:

Thank you for your precise comment. We rephrased that part of the text in order to clarify the treatment of the 2 groups as follows: “Treatment group A performed a rehabilitation treatment lasting 1 hour and 30 minutes of walk training on the Lokomat® plus a physiotherapy session consisting of a general exercise program and ground gait training; control group B performed only the physiotherapy treatment for 1 hour and 30 minutes.”

In addition, we explained what we mean by "conventional rehabilitation treatment": from line 224, we first changed the words "conventional rehabilitation treatment" to "conventional physiotherapy treatment", which consisted of a program of general exercises and ground gait training supervised by physiotherapist. A detailed explanation of the treatment is provided below in the text.

Comment 12: 158: “previously…stated/reported”

Response:

Thank you for your precise comments. We have edited the text accordingly.

Comment 13: 180: Report the sample size as ‘a priori’ or ‘post hoc’. I would assume a priori, but best to be clear.

Response:

Thank you for your precise comments. We have edited the text including “a priori”.

Comment 14: 183: an image/figure of the exoskeleton would be of interest to the reader, or a link to where this can be viewed.  

Response:

Thank you for your interesting idea. We entered in the text the following reference: “Riener, R.Technology of the robotic gait orthosis Lokomat (2016) Neurorehabilitation Technology, Second Edition, pp. 395-407”. In this study, the reader can view the images and find all the necessary information about the exoskeleton and its use.

Comment 15: 269-272: The is a small section of the written text dedicated to the non-significant differences and their p-values, can this be included in a table somewhere? Or, are these important to report at this level of detail.

Response:

We agree with the reviewer that this part is not very important to report all non-significant p values. We summarized the sentence as follows: “non-significant differences between T0 and T2 showed that the washout phase was effective for all outcomes (all p values > 0.12).”

Comment 16: 291: report the qualitative thresholds for the chosen effect size (i.e. small, medium, large), with a reference.

Response:

Thanks for this request. We included the qualitative thresholds of the effect sizes and the reference for that: “Cohen, J. (1988). Statistical Power Analysis for the Behavioral Sciences (2nd ed.). Routledge. https://doi.org/10.4324/9780203771587”. We also included the effect size in table 2.

Comment 17: Table 1: What is the 3rd Category – Average Type of Multiple Sclerosis = 59 = 50.4%. What is this? A reader would not know what 50.4% means.

Response:

We completely agree with the reviewer that the 3rd category was useless and misleading. We removed the 3rd column from the table. Thanks for spotting this.

Comment 18: Figure 3: does this figure provide the reader with anything beyond the written report?

Response:

Thanks for the comment. As the previous figure 3 did not add very much the report we decided to deleted it.

Comment 19: 114: This project was completed 6 year ago(?). Has the introduction/literature been updated since this project to reflect the most up to date evidence?

Response:

Thank you for your precise observation. Yes, we have updated both the introduction and the discussion with the most recent literature on the topic. For example, in the introduction, references number 3,4,5,14,17,21,22,23 included studies published from 2018 to 2021, and in particular a systematic review and meta-analysis on the effectiveness of training of robot-assisted gait in multiple sclerosis published in 2020 (22). We have also included some important recent studies in the discussion, for example references 53 to 56.

Comment 20: Data analysis and interpretation: From a clinical perspective, some interpretation of the results in terms of the clinical importance/significance of the noted changes would be of interest. The concept of the ‘smallest worthwhile change’ or ‘minimal clinically significant difference (MCID)’ could be applied for offer this. Based on the number of clinical outcome measures reported – which should be praised – the MCID or alike would offer a clinically meaningful assessment of these findings, in addition to the common/standard reliance of p-values.

Response:

We agree with the reviewer that including the minimal clinical important difference would be useful for clinicians to understand if the study results are clinically relevant. For this reason, we included in the first paragraph of the discussion a sentence regarding the primary outcome measure: “The average improvement in 25FW (14.9%) in the intervention group was larger than the minimal clinical important difference (Jensen et al 2016), therefore suggesting a clinically relevant enhancement in mobility.”

Comment 21: Additionally, some cost-based assessment could be of interest, i.e. what was the cost of delivering both interventions - clinicians would be interested in this. However, I will default to the journal editor to decide of this comment is in keeping with the Journal and its target audience.

Response:

Thanks for the observation. Our study did not aim to perform an analysis of the costs related to the type of treatment but only to evaluate its effectiveness. The use of the RAGT certainly involves significant costs linked to the purchase and management of the exoskeleton; on the other hand, it allows us to reduce the assistance time of physiotherapists directly engaged on the patient, in particular in patients who would need the assistance of 2 operators to perform gait training (with the exoskeleton one operator can manage the treatment). However, because the cost-benefit analysis is not the purpose of the study, it is not possible for us to provide more detailed information.

However, given the importance and interest that this aspect may arouse in the reader, we have included the following in the text: “The use of RAGT certainly entails significant costs linked to the purchase and management of the exoskeleton; on the other hand, it allows to reduce the number of physiotherapists directly engaged on the patient: one physiotherapist is able to manage the treatment with the exoskeleton, whereas 2 operators are usually required to perform gait training. Anyway, our study was not focused to perform a comparative cost-benefit analysis, but only to evaluate the effectiveness of two different treatment strategies. Nevertheless, this aspect is relevant in terms of Health policies and requires further investigations to estab-lish if RAGT might be economically rewarding.”
